# Noise Regularization for Conditional Density Estimation

## Abstract

Modelling statistical relationships beyond the conditional mean is crucial in many settings. *Conditional density estimation (CDE)* aims to learn the full conditional probability density from data. Though highly expressive, neural network based CDE models can suffer from severe over-fitting when trained with the maximum likelihood objective. Due to the inherent structure of such models, classical regularization approaches in the parameter space are rendered ineffective. To address this issue, we develop a *model-agnostic* noise regularization method for CDE that adds random perturbations to the data during training. We demonstrate that the proposed approach corresponds to a smoothness regularization and prove its asymptotic consistency. In our experiments, noise regularization significantly and consistently outperforms other regularization methods across seven data sets and three CDE models. The effectiveness of noise regularization makes neural network based CDE the preferable method over previous non- and semi-parametric approaches, even when training data is scarce.

## 1 Introduction

While regression analysis aims to describe the conditional mean $\mathbb{E}[y|x]$ of a response $y$ given inputs $x$, many problems such as risk management and planning under uncertainty require gaining insight about *deviations* from the mean and their associated likelihood. The stochastic dependency of $y$ on $x$ can be captured by modeling the conditional probability density $p(y|x)$. Inferring such a density function from a set of empirical observations $\{(x_n, y_n)\}_{n=1}^N$ is typically referred to as *conditional density estimation (CDE)* and is the focus of this paper.

In the recent machine learning literature, there has been a resurgence of interest in high-capacity density models based on neural networks (Dinh et al., 2017; Ambrogioni et al., 2017; Kingma & Dhariwal, 2018). Since this line of work mainly focuses on the modelling of images based on large scale data sets, over-fitting and noisy observations are of minor concern in this context. In contrast, we are interested in CDE in settings where data may be *scarce and noisy*. When combined with maximum likelihood estimation, the flexibility of such high-capacity models results in over-fitting and poor generalization. While regression typically assumes Gaussian conditional noise, CDE uses expressive distribution families to model deviations from the conditional mean. Hence, the over-fitting problem tends to be even more severe in CDE than in regression.

Classical regularization of the neural network weights such as weight decay (Pratt & Hanson, 1989) has been shown to be effective for regression and classification. However, in the context of CDE, the output of the neural network merely controls the parameters of a density model such as a Gaussian Mixture or Normalizing Flow. This makes the standard regularization methods in the parameter space less effective and harder to analyze.

Aiming to address this issue, we propose and analyze noise regularization, a method well-studied in the context of regression and classification, for the purpose of conditional density estimation. In that, the paper attempts to close a gap in previous research. By adding small random perturbations to the data during training, the conditional density estimate is smoothed and tends to generalize better. In fact, we show that adding noise during maximum likelihood estimation is equivalent to penalizing the second derivatives of the conditional log-probability. Visually, the respective regularization term punishes very curved or even spiky density estimators in favor of smoother variants, which proves to be a favorable inductive bias in many applications. Moreover, under some regularity conditions, we show that the proposed regularization scheme is asymptotically *consistent*, converging to the unbiased maximum likelihood estimator. This does not only support the soundness of the proposed

method but also endows us with useful insight in how to set the regularization intensity relative to the data dimensionality and training set size.

Overall, the proposed noise regularization scheme is easy to implement and agnostic to the parameterization of the CDE model. We empirically demonstrate its effectiveness on three different neural network based models. The experimental results show that noise regularization outperforms other regularization methods significantly and consistently across various data sets. Finally, we demonstrate that, when properly regularized, neural network based CDE is able to improve upon state-of-the art non-parametric estimators, even when only 400 training observations are available.

## 2  BACKGROUND

**Density Estimation.**   Let $X$ be a random variable with probability density function (PDF) $p(x)$ defined over the domain $\mathcal{X} \subseteq \mathbb{R}^{d_x}$. Given a collection $\mathcal{D} = \{x_1, ..., x_n\}$ of observations sampled from $p(x)$, the goal is to find a good estimate $\hat{f}(x)$ of the true density function $p$. In **parametric estimation**, the PDF $\hat{f}$ is assumed to belong to a parametric family $\mathcal{F} = \{\hat{f}_\theta(\cdot)|\theta \in \Theta\}$ where the density function is described by a finite dimensional parameter $\theta \in \Theta$. The standard method for estimating $\theta$ is *maximum likelihood estimation*, wherein $\theta$ is chosen so that the likelihood of the data $\mathcal{D}$ is maximized. This is equivalent to minimizing the Kullback-Leibler divergence between the empirical data distribution $p_\mathcal{D}(x) = \frac{1}{n}\sum_{i=1}^n \delta(||x - x_i||)$ (i.e., mixture of point masses in the observations $x_i$) and the parametric distribution $\hat{f}_\theta$:

$$\theta^* = \arg\max_{\theta \in \Theta} \sum_{i=1}^n \log \hat{f}_\theta(x_i) = \arg\min_{\theta \in \Theta} \mathcal{D}_{KL}(p_\mathcal{D}||\hat{f}_\theta) \tag{1}$$

From a geometric perspective, (1) can be viewed as an orthogonal projection of $p_\mathcal{D}(x)$ onto $\mathcal{F}$ w.r.t. the reverse KL-divergence. Hence, (1) is also commonly referred to as an *M-projection* (Murphy, 2012; Nielsen, 2018). In contrast, **non-parametric density estimators** make implicit smoothness assumptions through a kernel function. The most popular non-parametric method, kernel density estimation (KDE), places a symmetric density function $K(z)$, the so-called kernel, on each training data point $x_n$ (Rosenblatt, 1956; Parzen, 1962). The resulting density estimate reads as $\hat{q}(x) = \frac{1}{nh^d}\sum_{i=1}^n K\left(\frac{x-x_i}{h}\right)$. One popular choice of $K(\cdot)$ is a Gaussian $K(z) = (2\pi)^{-\frac{d}{2}}\exp\left(-\frac{1}{2}z^2\right)$. Beyond the appropriate choice of $K(\cdot)$, a central challenge is the selection of the bandwidth parameter $h$ which controls the smoothness of the estimated PDF (Li & Racine, 2007).

**Conditional Density Estimation (CDE).**   Let $(X, Y)$ be a pair of random variables with respective domains $\mathcal{X} \subseteq \mathbb{R}^{d_x}$ and $\mathcal{Y} \subseteq \mathbb{R}^{d_y}$ and realizations $x$ and $y$. Let $p(y|x) = p(x, y)/p(x)$ denote the conditional probability density of $y$ given $x$. Typically, $Y$ is referred to as a dependent variable (explained variable) and $X$ as conditional (explanatory) variable. Given a dataset of observations $\mathcal{D} = \{(x_n, y_n)\}_{n=1}^N$ drawn from the joint distribution $(x_n, y_n) \sim p(x, y)$, the aim of conditional density estimation (CDE) is to find an estimate $\hat{f}(y|x)$ of the true conditional density $p(y|x)$.

In the context of CDE, the KL-divergence objective is expressed as expectation over $p(x)$:

$$\mathbb{E}_{x\sim p(x)}\left[\mathcal{D}_{KL}(p(y|x)||\hat{f}(y|x))\right] = \mathbb{E}_{(x,y)\sim p(x,y)}\left[\log p(y|x) - \log \hat{f}(y|x)\right] \tag{2}$$

Corresponding to (1), we refer to the minimization of (2) w.r.t. $\theta$ as *conditional M-projection*. Given a dataset $\mathcal{D}$ drawn i.i.d. from $p(x, y)$, the conditional MLE following from (2) can be stated as

$$\theta^* = \arg\min_\theta -\sum_{i=1}^n \log \hat{f}_\theta(y_i|x_i) \tag{3}$$

## 3  RELATED WORK

The first part of this section discusses relevant work in the field of CDE, focusing on high-capacity models that make little prior assumptions. The second part relates our approach to previous regularization and data augmentation methods.

**Non-parametric CDE.**   A vast body of literature in statistics and econometrics studies nonparametric kernel density estimators (KDE) (Rosenblatt, 1956; Parzen, 1962) and the associated bandwidth selection problem, which concerns choosing the appropriate amount of smoothing (Silverman,

1982; Hall et al., 1992; Cao et al., 1994). To estimate conditional probabilities, previous work proposes to estimate both the joint and marginal probability separately with KDE and then computing the conditional probability as their ratio (Hyndman et al., 1996; Li & Racine, 2007). Other approaches combine non-parametric elements with parametric elements (Tresp, 2001; Sugiyama & Takeuchi, 2010; Dutordoir et al., 2018). Despite their theoretical appeal, non-parametric density estimators suffer from poor generalization in regions where data is sparse (e.g., tail regions), causing rapid performance deterioration as the data dimensionality increases (Scott & Wand, 1991).

**CDE based on neural networks.** Most work in machine learning focuses on flexible parametric function approximators for CDE. In our experiments, we use the work of Bishop (1994) and Ambrogioni et al. (2017), who propose to use a neural network to control the parameters of a mixture density model. A recent trend in machine learning are latent density models such as cGANs (Mirza & Osindero, 2014) and cVAEs (Sohn et al., 2015). Although such methods have been shown successful for estimating distributions of images, the probability density function (PDF) of such models is intractable. More promising in this sense are normalizing flows (Rezende & Mohamed, 2015; Dinh et al., 2017; Trippe & Turner, 2018), since they provide the PDF in tractable form. We employ a neural network controlling the parameters of a normalizing flow as our third CDE model to showcase the empirical efficacy of our regularization approach.

**Regularization.** Since neural network based CDE models suffer from severe over-fitting when trained with the MLE objective, they require proper regularization. Classical regularization of the parameters such as weight decay (Pratt & Hanson, 1989; Krogh & Hertz, 1992; Nowlan & Hinton, 1992), $l_1/l_2$-penalties (Mackay, 1992; Ng, 2004) and Bayesian priors (Murray & Edwards, 1993; Hinton & Van Camp, 1993) have been shown to work well in the regression and classification setting. However, in the context of CDE, it is less clear what kind of inductive bias such a regularization imposes on the density estimate. In contrast, our regularization approach is agnostic w.r.t. parametrization and is shown to penalize strong variations of the log-density function.

Regularization methods such as dropout are closely related to ensemble methods (Srivastava et al., 2014). Thus, they are orthogonal to our work and can be freely combined with noise regularization.

**Adding noise during training.** Adding noise during training is a common scheme that has been proposed in various forms. This includes noise on the neural network weights or activations (Wan et al., 2013; Srivastava et al., 2014; Gal & Uk, 2016) and additive noise on the gradients for scalable MCMC posterior inference (Welling & Teh, 2011; Chen et al., 2014). While this line of work corresponds to noise in the parameter space, other research suggests to augment the training data through random and/or adversarial transformations of the data (Sietsma & Dow, 1991; Burges & Schölkopf, 1996; Goodfellow et al., 2015; Yuan et al., 2017). Our approach transforms the training observations by adding small random perturbations. While this form of regularization has been studied in the context of regression and classification problems (Holmstrom & Koistinen, 1992a; Webb, 1994; Bishop, 1995; Natarajan et al., 2013; Maaten et al., 2013), this paper focuses on the regularization of CDE. In particular, we build on top of the results of Webb (1994) showing that training with noise corresponds to a penalty on strong variations of the log-density and extend previous consistency results for regression of Holmstrom & Koistinen (1992a) to the more general setting of CDE. To our best knowledge, this is also the first paper to evaluate the empirical efficacy of noise regularization for density estimation.

## 4 NOISE REGULARIZATION

When considering expressive families of conditional densities, standard maximum likelihood estimation of the model parameters $\theta$ is ill suited. As can be observed in Figure 1, simply minimizing the negative log-likelihood of the data leads to severe over-fitting and poor generalization beyond the training data. Hence, it is necessary to impose additional inductive bias, for instance, in the form of regularization. Unlike in regression or classification, the form of inductive bias imposed by popular regularization techniques such as weight decay (Krogh & Hertz, 1991; Kukačka et al., 2017) is less clear in the CDE setting, where the neural network weights often only indirectly control the probability density through a unconditional density model, e.g., a Gaussian Mixture.

We propose to add noise perturbations to the data points during the optimization of the log-likelihood objective. This can be understood as replacing the original data points $(x_i, y_i)$ by random variables $\tilde{x}_i = x_i + \xi_x$ and $\tilde{y}_i = y_i + \xi_y$ where the perturbation vectors are sampled from noise distribu-

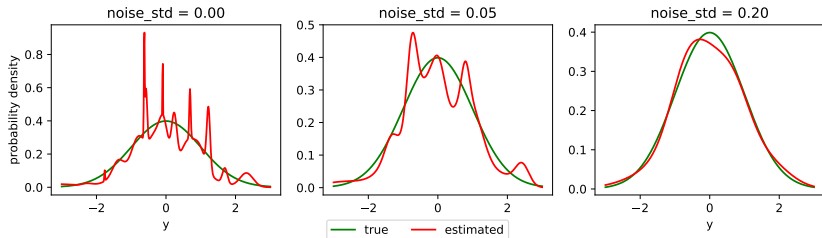

Figure 1: Conditional MDN density estimate (red) and true conditional density (green) for different noise regularization intensities $h_z \in \{0.0, 0.05, 0.2\}$. The MDN has been fitted with 3000 samples drawn from a conditional Gaussian.

tions $K_x(\xi_x)$ and $K_y(\xi_y)$ respectively. Further, we choose the noise to be zero centered as well as identically and independently distributed among the data dimensions, with standard deviation $h$:

$$\mathbb{E}_{\xi \sim K(\xi)}[\xi] = 0 \quad \text{and} \quad \mathbb{E}_{\xi \sim K(\xi)}[\xi \xi^\top] = h^2 I \tag{4}$$

This can be seen as data augmentation, where "synthetic" data is generated by randomly perturbing the original data. Since the supply of noise vectors is technically unlimited, an arbitrary large augmented data set can be generated by repetitively sampling data points from $\mathcal{D}$, and adding a random perturbation vector to the respective data point. This procedure is formalized in Algorithm 1.

For notational brevity, we set $\mathcal{Z} := \mathcal{X} \times \mathcal{Y}$, $z := (x^\top, y^\top)^\top$ and denote $\hat{f}_\theta(z) := \hat{f}_\theta(y|x)$. The presented noise regularization approach is agnostic to whether we are concerned with unconditional or conditional MLE. Thus, the generic notation also allows us to generalize the results to both settings (derived in the remainder of the paper).

---

**Algorithm 1** (Conditional) MLE with Noise Regularization - Generic Procedure

---

**Require:** $\mathcal{D} = \{z_1, ..., z_n\}$, noise intensity $h$
**Require:** number of perturbed samples $r$,
 1: **for** j = 1 to r **do**
 2:     Select $i \in \{1, ..., n\}$ with equal prob.
 3:     Draw perturbation $\xi \sim K$
 4:     Set $\tilde{z}_j = z_i + h\xi$
 5: **return** $\arg\min_{\theta \in \Theta} - \sum_{j=1}^r \log \hat{f}_\theta(\tilde{z}_j)$

---

**Algorithm 2** (Conditional) MLE with Noise Regularization - Mini-Batch Gradient Descent

---

**Require:** $\mathcal{D} = \{z_1, ..., z_n\}$, noise intensity $h$
**Require:** learning rate $\alpha$, mini-batch size $m$
 1: Initialize $\theta$
 2: **while** $\theta$ not converged **do**
 3:     Sample minibatch $\{z_1, ..., z_m\} \subset \mathcal{D}$
 4:     **for** j = 1 to m **do**
 5:         Draw perturbation $\xi \sim K$
 6:         Set $\tilde{z}_j = z_j + h\xi$
 7:     $\theta \leftarrow \theta + \alpha \nabla_\theta \sum_{j=1}^m \log \hat{f}_\theta(\tilde{z}_j)$
 8: **return** optimized parameter $\theta$

---

When considering highly flexible parametric families such as Mixture Density Networks (MDNs) (Bishop, 1994), the maximum likelihood solution in line 5 of Algorithm 1 is no longer tractable. In such case, one typically resorts to numerical optimization techniques such as mini-batch gradient descent and variations thereof. In this context, the generic procedure in Algorithm 1 can be transformed into a simple extensions of mini-batch gradient descent on the MLE objective (see Algorithm 2). Specifically, each mini-batch is perturbed with i.i.d. noise before computing the MLE objective function (forward pass) and the respective gradients (backward pass).

### 4.1 VARIABLE NOISE AS SMOOTHNESS REGULARIZATION

Intuitively, the previously presented variable noise can be interpreted as "smearing" the data points during the maximum likelihood estimation. This alleviates the jaggedness of the density estimate arising from an un-regularized maximum likelihood objective in flexible density classes. We will now give this intuition a formal foundation, by mathematically analyzing the effect of the noise perturbations.

Before discussing the particular effects of randomly perturbing the data during conditional maximum likelihood estimation, we first analyze noise regularization in a more general case. Let $l_(\mathcal{D})$ be a loss function over a set of data points $\mathcal{D} = \{z_1, ..., z_n\}$, which can be partitioned into a sum of losses $l(\mathcal{D}) = \sum_{i=1}^n l(z_i)$, corresponding to each data point $z_i$: The expected loss $l(z_i + \xi)$, resulting from adding random perturbations, can be approximated by a second order Taylor expansion around

$z_i$. Using the assumption about $\xi$ in (4), the expected loss an be written as

$$\mathbb{E}_{\xi \sim K(\xi)}\left[l(z_i + \xi)\right] = l(z_i) + \frac{1}{2}\mathbb{E}_{\xi \sim K(\xi)}\left[\xi^\top \mathbf{H}^{(i)}\xi\right] + \mathcal{O}(\xi^3) \approx l(z_i) + \frac{h^2}{2}\text{tr}(\mathbf{H}^{(i)}) \qquad (5)$$

where $l(\mathbf{z}_i)$ is the loss without noise and $\mathbf{H}^{(i)} = \frac{\partial^2 l}{\partial z^2}(z)\big|_{z_i}$ the Hessian of $l$ w.r.t $z$, evaluated at $z_i$. Assuming that the noise $\xi$ is small in its magnitude, $\mathcal{O}(\xi^3)$ is negligible. This effect has been observed earlier by Webb (1994) and Bishop (1994). See Appendix A for derivations.

When concerned with maximum likelihood estimation of a conditional density $\hat{f}_\theta(y|x)$, the loss function coincides with the negative conditional log-likelihood $l(y_i, x_i) = -\log \hat{f}_\theta(y_i|x_i)$. Let the standard deviation of the additive data noise $\xi_x, \xi_y$ be $h_x$ and $h_y$ respectively. Maximum likelihood estimation (MLE) with data noise is equivalent to minimizing the loss

$$l(\mathcal{D}) \approx -\sum_{i=1}^{n}\log \hat{f}_\theta(y_i|x_i) - \frac{h_y^2}{2}\sum_{i=1}^{n}\sum_{j=1}^{d_y}\frac{\partial^2 \log \hat{f}_\theta(y|x)}{\partial y^{(j)}\partial y^{(j)}}\bigg|_{\substack{x=x_i \\ y=y_i}} - \frac{h_x^2}{2}\sum_{i=1}^{n}\sum_{j=1}^{d_x}\frac{\partial^2 \log \hat{f}_\theta(y|x)}{\partial x^{(j)}\partial x^{(j)}}\bigg|_{\substack{x=x_i \\ y=y_i}} \qquad (6)$$

In that, the first term corresponds to the standard MLE objective, while the other two terms constitute a form of smoothness regularization. The second term in (6) penalizes large negative second derivatives of the conditional log density estimate $\log \hat{f}_\theta(y|x)$ w.r.t. $y$. As the MLE objective pushes the density estimate towards high densities and strong concavity in the data points $y_i$, the regularization term counteracts this tendency to over-fit and overall smoothes the fitted distribution. The third term penalizes large negative second derivatives w.r.t. the conditional variable $x$, thereby regularizing the sensitivity of the density estimate to changes in the conditional variable. The intensity of the noise regularization can be controlled through the variance ($h_x^2$ and $h_y^2$) of the random perturbations.

Figure 1 illustrates the effect of the introduced noise regularization scheme on MDN estimates. Plain maximum likelihood estimation (left) leads to strong over-fitting, resulting in a spiky distribution that generalizes poorly beyond the training data. In contrast, training with noise regularization (center and right) results in smoother density estimates that are closer to the true conditional density.

## 4.2 CONSISTENCY OF NOISE REGULARIZATION

We now establish asymptotic consistency results for the proposed noise regularization. In particular, we show that, under some regularity conditions, concerning integrability and decay of the noise regularization, the solution of Algorithm 1 converges to the asymptotic MLE solution.

Let $\hat{f}_\theta(z) : \mathbb{R}^{d_z} \times \Theta \to (0, \infty)$ a continuous function of $z$ and $\theta$. Moreover, we assume that the parameter space $\Theta$ is compact. In the classical MLE setting, the idealized loss, corresponding to a (conditional) M-projection of the true data distribution onto the parametric family, reads as

$$l(\theta) = -\mathbb{E}_{p(z)}\left[\log \hat{f}_\theta(z)\right] \qquad (7)$$

As we typically just have a finite number of samples from $p(z)$, the respective empirical estimate $\hat{l}_n(\theta) = -\frac{1}{n}\sum_{i=1}^{n}\log \hat{f}_\theta(z_i)$, $z_i \overset{i.i.d}{\sim} p(z)$ is used as training objective. Note that we now define the loss as function of $\theta$, and, for fixed $\theta$, treat $l_n(\theta)$ as a random variable. Under some regularity conditions, one can invoke the uniform law of large numbers to show consistency of the empirical ML objective in the sense that $\sup_{\theta \in \Theta}|\hat{l}_n(\theta) - l(\theta)| \xrightarrow{a.s.} 0$ (see Appendix B for details).

In case of the presented noise regularization scheme, the maximum likelihood estimation is performed using on the augmented data $\{\tilde{z}_j\}$ rather than the original data $\{z_i\}$. For our analysis, we view Algorithm 1 from a slightly different angle. In fact, the data augmentation procedure of uniformly selecting a data point from $\{z_1, ..., z_n\}$ and perturbing it with a noise vector drawn from $K$ can be viewed as drawing i.i.d. samples from a kernel density estimate $\hat{q}_n^{(h)}(z) = \frac{1}{n}\sum_{i=1}^{n}\frac{1}{h^{d_z}}K\left(\frac{z-z_i}{h}\right)$. Hence, maximum likelihood estimation with variable noise can be understood as

1. forming a kernel density estimate $\hat{q}_n^{(h)}$ of the training data

2. followed by a (conditional) M-projection of $\hat{q}_n^{(h)}$ onto the parametric family.

In that, step 2 aims to find the $\theta^*$ that minimizes the following objective:

$$l_n^{(h)}(\theta) = -\mathbb{E}_{\hat{q}_n^{(h)}(z)}\left[\log \hat{f}_\theta(z)\right] \tag{8}$$

Since (8) is generally intractable, $r$ samples are drawn from the kernel density estimate, forming the following Monte Carlo approximation of (8) which corresponds to the loss in line 5 Algorithm 1:

$$\hat{l}_{n,r}^{(h)}(\theta) = -\frac{1}{r}\sum_{j=1}^{r}\log \hat{f}_\theta(\tilde{z}_j), \quad \tilde{z}_j \sim \hat{q}_n^{(h)}(z) \tag{9}$$

We are concerned with the consistency of the training procedure in Algorithm 1, similar to the classical MLE consistency result discussed above. Hence, we need to show that $\sup_{\theta\in\Theta}\left|\hat{l}_{n,r}^{(h)}(\theta) - l(\theta)\right| \xrightarrow{a.s.} 0$ as $n, r \to \infty$. We begin our argument by decomposing the problem into easier sub-problems. In particular, the triangle inequality is used to obtain the following upper bound:

$$\sup_{\theta\in\Theta}\left|\hat{l}_{n,r}^{(h)}(\theta) - l(\theta)\right| \leq \sup_{\theta\in\Theta}\left|\hat{l}_{n,r}^{(h)}(\theta) - l_n^{(h)}(\theta)\right| + \sup_{\theta\in\Theta}\left|l_n^{(h)}(\theta) - l(\theta)\right| \tag{10}$$

Note that $\hat{l}_{n,r}^{(h)}(\theta)$ is based on samples from the kernel density estimate, which are obtained by adding random noise vectors $\xi \sim K(\cdot)$ to our original training data. Since we can sample an unlimited amount of such random noise vectors, $r$ can be chosen arbitrarily high. This allows us to make $\sup_{\theta\in\Theta}|\hat{l}_{n,r}^{(h)}(\theta) - l_n^{(h)}(\theta)|$ arbitrary small by the uniform law of large numbers. In order to make $\sup_{\theta\in\Theta}|l_n^{(h)}(\theta) - l(\theta)|$ small in the limit $n \to \infty$, the sequence of bandwidth parameters $h_n$ needs to be chosen appropriately. Such results can then be combined using a union bound argument. In the following we outline the steps leading us to the desired results. In that, the proof methodology is similar to Holmstrom & Koistinen (1992b). While they show consistency results for regression with a quadratic loss function, our proof deals with generic and inherently unbounded log-likelihood objectives and thus holds for a much more general class of learning problems. The full proofs can be found in the Appendix.

Initially, we have to make asymptotic integrability assumptions that ensure that the expectations in $l_n^{(h)}(\theta)$ and $l(\theta)$ are well-behaved in the limit (see Appendix C for details). Given respective integrability, we are able to obtain the following proposition.

**Proposition 1** *Suppose the regularity conditions (28) and (29) are satisfied, and that*

$$\lim_{n\to\infty} h_n = 0, \qquad \lim_{n\to\infty} n(h_n)^d = \infty \tag{11}$$

*Then,*

$$\lim_{n\to\infty} \sup_{\theta\in\Theta}\left|l_n^{(h)}(\theta) - l(\theta)\right| = 0 \tag{12}$$

*almost surely.*

In (11) we find conditions on the asymptotic behavior of the smoothing sequence $(h_n)$. These conditions also give us valuable guidance on how to properly choose the noise intensity in line 4 of Algorithm 1 (see Section 4.3 for discussion). The result in (12) demonstrates that, under the discussed conditions, replacing the empirical data distribution with a kernel density estimate still results in an asymptotically consistent maximum likelihood objective. However, as previously discussed, $l_n^{(h)}(\theta)$ is intractable and, thus, replaced by its sample estimate $\hat{l}_{n,r}^{(h)}$. Since we can draw an arbitrary amount of samples from $\hat{q}_n^{(h)}$, we can approximate $l_n^{(h)}(\theta)$ with arbitrary precision. Given a fixed data set $\mathcal{D}$ of size $n > n_0$, this means that $\lim_{r\to\infty}\sup_{\theta\in\Theta}\left|\hat{l}_{n,r}^{(h)}(\theta) - l_n^{(h)}(\theta)\right| = 0$ almost surely, by (29) and the uniform law of large numbers. Since our original goal was to also show consistency for $n \to \infty$, this result is combined with Proposition 1, obtaining the following consistency theorem.

**Theorem 1** *Suppose the regularity conditions (28) and (29) are satisfied, $h_n$ fulfills (11) and $\Theta$ is compact. Then,*

$$\lim_{n\to\infty} \overline{\lim_{r\to\infty}} \sup_{\theta\in\Theta}\left|\hat{l}_{n,r}^{(h)}(\theta) - l(\theta)\right| = 0 \tag{13}$$

*almost surely.*

In that, $\overline{\lim}$ used to denote the limit superior ("lim sup") of a sequence.

Training a (conditional) density model with noise regularization means minimizing $\hat{l}_{n,r}^{(h)}(\theta)$ w.r.t. $\theta$. As result of this optimization, one obtains a parameter vector $\hat{\theta}_{n,r}^{(h)}$, which we hope is close to the minimizing parameter $\bar{\theta}$ of the ideal objective function $l(\theta)$. In the following, we establish asymptotic consistency results, similar to Theorem 1, in the parameter space. Therefore we first have to formalize the concept of closeness and optimality in the parameter space. Since a minimizing parameter $\bar{\theta}$ of $l(\theta)$ may not be unique, we define $\Theta^* = \{\theta^* \mid l(\theta^*) \le l(\theta) \, \forall \theta \in \Theta\}$ as the set of global minimizers of $l(\theta)$, and $d(\theta, \Theta^*) = \min_{\theta^* \in \Theta^*} \{||\theta - \theta^*||_2\}$ as the distance of an arbitrary parameter $\theta$ to $\Theta^*$. Based on these definitions, it can be shown that Algorithm 1 is asymptotically consistent in a sense that the minimizer of $\hat{\theta}_{n,r}^{(h)}$ converges almost surely to the set of optimal parameters $\Theta^*$.

**Theorem 2** *Suppose the regularity conditions (28) and (29) are satisfied, $h_n$ fulfills (11) and $\Theta$ is compact. For $r > 0$ and $n > n_0$, let $\hat{\theta}_{n,r}^{(h)} \in \Theta$ be a global minimizer of the empirical objective $\hat{l}_{n,r}^{(h)}$. Then*

$$\lim_{n \to \infty} \overline{\lim_{r \to \infty}} \, d(\hat{\theta}_{n,r}^{(h)}, \Theta^*) = 0 \tag{14}$$

*almost surely.*

Note that Theorem 2 considers global optimizers, but equivalently holds for compact neighborhoods of a local minimum $\theta^*$ (see discussion in Appendix C).

### 4.3 CHOOSING THE NOISE INTENSITY

After discussing the properties of noise regularization, we are interested in how to properly choose the noise intensity $h$, for different training data sets. Ideally, we would like to choose $h$ so that $|l_n^{(h)}(\theta) - l(\theta)|$ is minimized, which is practically not feasible since $l(\theta)$ is intractable. Inequality (30) gives as an upper bound on this quantity, suggesting to minimize $l_1$ distance between the kernel density estimate $q_n^{(h)}$ and the data distribution $p(z)$. This is in turn a well-studied problem in the kernel density estimation literature (see e.g., Devroye & Luc (1987)). Unfortunately, general solutions of this problem require knowing $p(z)$ which is not the case in practice. Under the assumption that $p(z)$ and the kernel function $K$ are Gaussian, the optimal bandwidth can be derived as $h = 1.06\hat{\sigma}n^{-\frac{1}{4+d}}$ (Silverman, 1986). In that, $\hat{\sigma}$ denotes the estimated standard deviation of the data, $n$ the number of data points and $d$ the dimensionality of $\mathcal{Z}$. This formula is widely known as the *rule of thumb* and often used as a heuristic for choosing $h$.

In addition, the conditions in (11) give us further intuition. The first condition tells us that $h_n$ needs to decay towards zero as $n$ becomes large. This reflects the general theme in machine learning that the more data is available, the less inductive bias / regularization should be imposed. The second condition suggests that the bandwidth decay must happen at a rate slower than $n^{-\frac{1}{d}}$. For instance, the rule of thumb fulfills these two criteria and thus constitutes a useful guideline for selecting $h$. However, for highly non-Gaussian data distributions, the respective $h_n$ may decay too slowly and a faster decay rate such as $n^{-\frac{1}{1+d}}$ may be appropriate.

## 5 EXPERIMENTS

This section provides a detailed experimental analysis of the proposed method, aiming to empirically validate the theoretical arguments outlined previously and investigating the practical efficacy of our regularization approach. In all experiments we use Gaussian pertubations of the data, i.e., $K(\xi) = \mathcal{N}(0, I)$. Since one of the key features of our noise regularization scheme is that it is agnostic to the choice of model, we evaluate its performance on three different neural network based CDE models: Mixture Density Networks (MDN) (Bishop, 1994), Kernel Mixture Networks (KMN) (Ambrogioni et al., 2017) and Normalizing Flows Networks (NFN) (Rezende & Mohamed, 2015; Trippe & Turner, 2018). In our experiments, we consider both simulated as well as real-world data sets. In particular, we simulate data from a 4-dimensional Gaussian Mixture ($d_x = 2, d_y = 2$) and a Skew-Normal distribution whose parameters are functionally dependent on $x$ ($d_x = 1, d_y = 1$). In terms of real-world data, we use the following three data sources:

**Euro Stoxx:** Daily stock-market returns of the Euro Stoxx 50 index conditioned on various stock return factors relevant in finance ($d_x = 14, d_y = 1$).

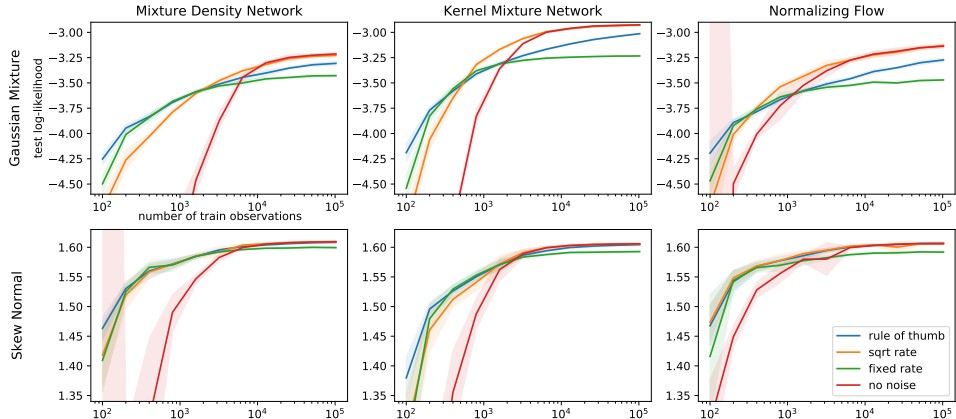

Figure 2: Comparison of different noise intensity schedules $h_n$ and their implications on the performance of various CDE models across different training set sizes.

**NYC Taxi:** Drop-off locations of Manhattan taxi trips conditioned on the pickup location, weekday and time ($d_x = 6, d_y = 2$).

**UCI datasets:** Standard data sets from the UCI machine learning repository (Dua & Graff, 2017). In particular, Boston Housing ($d_x = 13, d_y = 1$), Concrete ($d_x = 8, d_y = 1$), Energy ($d_x = 9, d_y = 1$).

The reported scores are test log-likelihoods, averaged over at least 5 random seeds alongside the respective standard deviation. For further details regarding the data sets and simulated data, we refer to Appendix E. The experiment data and code is available at TODO

### 5.1 NOISE INTENSITY SCHEDULES

We complement the discussion in 4.3 with an empirical investigation of different schedules of $h_n$. In particular, we compare a) the rule of thumb $h_n \propto n^{-\frac{1}{4+d}}$ b) a square root decay schedule $h_n \propto n^{-\frac{1}{1+d}}$ c) a constant bandwidth $h_n = const. \in (0, \infty)$ and d) no noise regularization, i.e. $h_n = 0$. Figure 2 plots the respective test log-likelihoods against an increasing training set size $n$ for the two simulated densities Gaussian Mixture and Skew Normal.

First, we observe that bandwidth rates which conform with the decay conditions seem to converge in performance to the non-regularized maximum likelihood estimator (red) as $n$ becomes large. This reflects the theoretical result of Theorem 1. Second, a fixed bandwidth across $n$ (green), violating (11), imposes asymptotic bias and thus saturates in performance vastly before its counterparts. Third, as hypothesized, the relatively slow decay of $h_n$ through the rule of thumb works better for data distributions that have larger similarities to a Gaussian, i.e., in our case the Skew Normal distribution. In contrast, the highly non-Gaussian data from the Gaussian Mixture requires faster decay rates like the square root decay schedule. Most importantly, noise regularization substantially improves the estimator's performance when only little training data is available.

### 5.2 REGULARIZATION COMPARISON

We now investigate how the proposed noise regularization scheme compares to classical regularization techniques. In particular, we consider an $l_1$ and $l_2$-penalty on the neural network weights as regularization term, the weight decay technique of Loshchilov & Hutter (2019)[1], as well a Bayesian neural network (Neal, 2012) trained with variational inference using a Gaussian prior and posterior (Blei et al., 2017).

First, we study the performance of the regularization techniques on our two simulation benchmarks. Figure 3 depicts the respective test log-likelihood across different training set sizes. For each regularization method, the regularization hyper-parameter has been optimized via grid search.

As one would expect, the importance of regularization, i.e., performance difference to un-regularized model, decreases as the amount of training data becomes larger. The noise regularization scheme

---

[1] Note that an $l_2$ regularizer and weight decay are not equivalent since we use the adaptive learning rate technique Adam. See Loshchilov & Hutter (2019) for details.

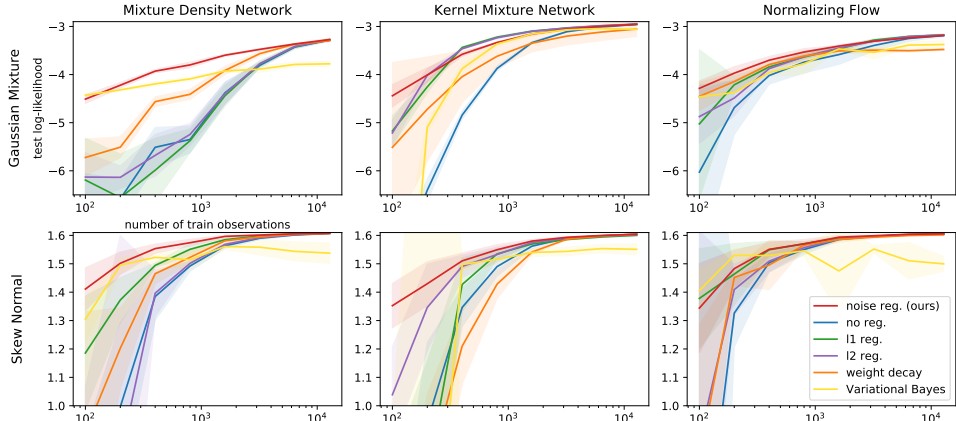

Figure 3: Comparison of various regularization methods for three neural network based CDE models. The models are trained with simulated data sets of different sizes.

|  |  | Euro Stoxx | NYC Taxi | Boston | Concrete | Energy |
|---|---|---|---|---|---|---|
| **MDN** | noise (ours) | **3.94±0.03** | **5.25±0.04** | **-2.49±0.11** | **-2.92±0.08** | **-1.04±0.09** |
|  | weight decay | 3.78±0.06 | 5.07±0.04 | -3.29±0.32 | -3.33±0.14 | -1.21±0.10 |
|  | l1 reg. | 3.19±0.19 | 5.00±0.05 | -4.01±0.36 | -3.87±0.29 | -1.44±0.22 |
|  | l2 reg. | 3.16±0.21 | 4.99±0.04 | -4.64±0.52 | -3.84±0.26 | -1.55±0.26 |
|  | Bayes | 3.26±0.43 | 5.08±0.03 | -3.46±0.47 | -3.19±0.21 | -1.25±0.23 |
| **KMN** | noise (ours) | **3.92±0.01** | **5.39±0.02** | **-2.52±0.08** | **-3.09±0.06** | **-1.62±0.06** |
|  | weight decay | 3.85±0.03 | 5.31±0.02 | -2.69±0.15 | -3.15±0.06 | -1.79±0.12 |
|  | l1 reg. | 3.76±0.04 | **5.39±0.02** | -2.75±0.13 | -3.25±0.07 | -1.82±0.10 |
|  | l2 reg. | 3.71±0.05 | 5.37±0.02 | -2.66±0.13 | -3.18±0.07 | -1.79±0.13 |
|  | Bayes | 3.33±0.02 | 4.47±0.02 | -3.40±0.11 | -4.08±0.05 | -3.65±0.07 |
| **NFN** | noise (ours) | **3.90±0.01** | **5.20±0.03** | **-2.48±0.11** | **-3.03±0.13** | -1.21±0.08 |
|  | weight decay | 3.82±0.06 | 5.19±0.03 | -3.12±0.39 | -3.12±0.14 | -1.22±0.16 |
|  | l1 reg. | 3.50±0.10 | 5.12±0.05 | -12.58±12.76 | -3.91±0.52 | -1.29±0.16 |
|  | l2 reg. | 3.50±0.09 | 5.13±0.05 | -14.22±9.60 | -3.99±0.66 | -1.34±0.19 |
|  | Bayes | 3.34±0.33 | 5.10±0.03 | -5.99±2.45 | -3.55±0.46 | **-1.11±0.22** |

Table 1: Comparison of various regularization methods for three neural network based CDE models across 5 data sets. We report the test log-likelihood and its respective standard deviation (higher log-likelihood values are better).

yields similar performance across the different CDE models while the other regularizers vary greatly in their performance depending on the different models. This reflects the fact that noise regularization is agnostic to the parameterization of the CDE model while regularizers in the parameter space are dependent on the internal structure of the model. Most importantly, noise regularization performs well across all models and sample sizes. In the great majority of configurations it outperforms the other methods. Especially when little training data is available, noise regularization ensures a moderate test error while the other approaches mostly fail to do so.

Next, we consider real world data sets. Since now the amount of data we can use for hyper-parameter selection, training and evaluation is limited, we use 5-fold cross-validation to select the parameters for each regularization method. The test log-likelihoods, reported in Table 1, are averages over 3 different train/test splits and 5 seeds each for initializing the neural networks. The held out test set amounts to 20% of the overall data sets. Consistent with the results of the simulation study, noise regularization outperforms the other methods across the great majority of data sets and CDE models.

### 5.3 CONDITIONAL DENSITY ESTIMATOR BENCHMARK STUDY

We benchmark neural network based density estimators against state-of-the art CDE approaches. While neural networks are the obvious choice when a large amount of training data is available, we pose the questions how such estimators compete against well-established non-parametric methods in small data regimes. In particular, we compare to the three following CDE methods:

**Conditional Kernel Density Estimation (CKDE).** Non-parametric method that forms a KDE of both $p(x, y)$ and $p(x)$ to compute its estimate as $\hat{p}(y|x) := \hat{p}(x, y)/\hat{p}(x)$ (Li & Racine, 2007).

|  | Euro Stoxx | NCY Taxi | Boston | Conrete | Energy |
|---|---|---|---|---|---|
| num. train obs. | 2536 | 8000 | 405 | 824 | 615 |
| MDN | **4.00±0.03** | 5.41±0.02 | **-2.39±0.02** | **-2.89±0.03** | **-1.04±0.05** |
| KMN | 3.98±0.03 | **5.42±0.02** | -2.44±0.02 | -3.06±0.03 | -1.59±0.09 |
| NFN | **4.00±0.03** | 5.12±0.03 | -2.40±0.04 | -2.93±0.02 | -1.23±0.06 |
| LSCDE | 3.44±0.10 | 4.85±0.02 | -2.78±0.00 | -3.63±0.00 | -2.16±0.02 |
| CKDE R.O.T. | 3.36±0.01 | 4.87±0.02 | -3.12±0.03 | -3.78±0.02 | -2.90±0.01 |
| CKDE CV-ML | 3.87±0.01 | 5.27±0.06 | -2.76±0.26 | -3.35±0.13 | -1.14±0.02 |
| NKDE R.O.T | 3.16±0.02 | 4.34±0.04 | -3.52±0.05 | -4.08±0.02 | -3.35±0.03 |
| NKDE CV-ML | 3.41±0.02 | 4.93±0.08 | -3.34±0.13 | -3.93±0.05 | -2.21±0.12 |

Table 2: Comparison of conditional density estimators across 5 data sets. Reported is the test log-likelihood and its respective standard deviation (higher log-likelihood values are better).

**$\epsilon$-Neighborhood kernel density estimation (NKDE).** Non-parametric method that considers only a local subset of training points to form a density estimate.

**Least-Squares Conditional Density Estimation (LSCDE).** Semi-parametric estimator that computes the conditional density as linear combination of fixed kernels (Sugiyama & Takeuchi, 2010).

For the kernel density estimation based methods CKDE and NKDE, we perform bandwidth selection via the rule of thumb (R.O.T) (Silverman, 1982; Sheather & Jones, 1991) and via maximum likelihood leave-one-out cross-validation (CV-ML) (Rudemo, 1982; Hall et al., 1992). In case of LSCDE, MDN, KMN and NFN, the respective hyper-parameters are selected via 5-fold cross-validation grid search on the training set. Note that, in contrast to Section 5.2 which focuses on regularization parameters, the grid search here extends to more hyper-parameters. The respective test log-likelihood scores are listed in Table 2. For the majority of data sets, all three neural network based methods outperform all of the non- and semi-parametric methods. Perhaps surprisingly, it can be seen that, when properly regularized, neural network based CDE works well even when training data is *scarce*, such as in case of the Boston Housing data set.

## 6 CONCLUSION

This paper addresses conditional density estimation with high-capacity models. In particular, we propose to add small random perturbations to the data during training. We demonstrate that the resulting noise regularization method corresponds to a smoothness regularization and prove its asymptotic consistency. The experimental results underline the effectiveness of the proposed method, demonstrating that it consistently outperforms other regularization methods across various conditional density models and data sets. This makes neural network based CDE the preferable method, even when only little training data is available. While we assess the estimator performance in terms of the test log-likelihood, an interesting question for future research is whether the noise regularization also improves the respective uncertainty estimates for downstream tasks such as safe control and decision making.

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

## A   DERIVATION SMOOTHNESS REGULARIZATION

Let $l(\mathcal{D})$ be a loss function over a set of data points $\mathcal{D} = \{z_1, ..., z_N\}$, which can be partitioned into a sum of losses corresponding to each data point $x_n$:

$$l_\mathcal{D}(\mathcal{D}) = \sum_{i=1}^{n} l(z_i) \tag{15}$$

Also, let each $z_i$ be perturbed by a random noise vector $\xi \sim K(\xi)$ with zero mean and i.i.d. elements, i.e.

$$\mathbb{E}_{\xi \sim K(\xi)}\left[\xi\right] = 0 \quad \text{and} \quad \mathbb{E}_{\xi \sim K(\xi)}\left[\xi_n \xi_j^\top\right] = h^2 I \tag{16}$$

The resulting loss $l(z_i + \xi)$ can be approximated by a second order Taylor expansion around $z_i$

$$l(z_i + \xi) = l(z_i) + \xi^\top \nabla_z l(z)\big|_{z_i} + \frac{1}{2}\xi^\top \nabla_z^2 l(z)\big|_{z_i}\xi + \mathcal{O}(\xi^3) \tag{17}$$

Assuming that the noise $\xi$ is small in its magnitude, $\mathcal{O}(\xi^3)$ may be neglected. The expected loss under $K(\xi)$ follows directly from (17):

$$\mathbb{E}_{\xi \sim K(\xi)}\left[l(z_i + \xi)\right] = l(z_i) + \mathbb{E}_{\xi \sim K(\xi)}\left[\xi^\top \nabla_x l(z)\big|_{z_i}\right] + \frac{1}{2}\mathbb{E}_{\xi \sim K(\xi)}\left[\xi^\top \nabla_x^2 l(z)\big|_{z_i}\xi\right] \tag{18}$$

Using the assumption about $\xi$ in (16) we can simplify (18) as follows:

$$\mathbb{E}_{\xi \sim K(\xi)}\left[l(z_i + \xi)\right] = l(z_i) + \mathbb{E}_{\xi \sim K(\xi)}\left[\xi\right]^\top \nabla_z l(z)\big|_{z_i} + \frac{1}{2}\mathbb{E}_{\xi \sim K(\xi)}\left[\xi^\top \nabla_z^2 l(z)\big|_{z_i}\xi\right] \tag{19}$$

$$= l(z_i) + \frac{1}{2}\mathbb{E}_{\xi \sim K(\xi)}\left[\xi^\top \mathbf{H}^{(i)}\xi\right] \tag{20}$$

$$= l(z_i) + \frac{1}{2}\mathbb{E}_{\xi \sim K(\xi)}\left[\sum_j \sum_k \xi_j \xi_k \frac{\partial^2 l(z)}{\partial z^{(j)}\partial z^{(k)}}\bigg|_{z_i}\right] \tag{21}$$

$$= l(z_i) + \frac{1}{2}\sum_j \mathbb{E}_\xi\left[\xi_j^2\right]\frac{\partial^2 l(z)}{\partial z^{(j)}\partial z^{(j)}}\bigg|_{z_i} + \frac{1}{2}\sum_j \sum_{k \neq j}\mathbb{E}_\xi\left[\xi_j \xi_k\right]\frac{\partial^2 l(z)}{\partial z^{(j)}\partial z^{(k)}}\bigg|_{z_i} \tag{22}$$

$$= l(z_i) + \frac{\eta^2}{2}\sum_j \frac{\partial^2 l(z)}{\partial z^{(j)}\partial z^{(j)}}\bigg|_{z_i} \tag{23}$$

$$= l(z_i) + \frac{\eta^2}{2}\text{tr}(\mathbf{H}^{(i)}) \tag{24}$$

In that, $l(z_i)$ is the loss without noise and $\mathbf{H}^{(i)} = \nabla_z^2 l(z)\big|_{z_i}$ the Hessian of $l$ at $z_i$. With $z^{(j)}$ we denote the elements of the column vector $z$.

## B   VANILLA CONDITIONAL MLE OBJECTIVE IS UNIFORMLY CONSISTENT

The objective function corresponding to a conditional M-projection.

$$l(\theta) = -\mathbb{E}_{p(x,y)}\left[\log \hat{f}_\theta(y|x)\right] \tag{25}$$

The sample equivalent:

$$\hat{l}_n(\theta) = -\frac{1}{n}\sum_{i=1}^{n}\log \hat{f}_\theta(y_i|x_i), \quad (x_i, y_i) \overset{i.i.d}{\sim} P(X, Y) \tag{26}$$

**Corollary 1** *Let $\Theta$ be a compact set and and $\hat{f}_\theta : \mathbb{R}^l \times \mathbb{R}^m \times \Theta \to (0, \infty)$ continuous in $\theta$ for all $(x, y) \in \mathbb{R}^l \times \mathbb{R}^m$ such that $\mathbb{E}_{p(x,y)}\left[\sup_{\theta \in \Theta}\log \hat{f}_\theta(y|x)\right] < \infty$. Then, as $n \to \infty$, we have*

$$\sup_{\theta \in \Theta}\left|\hat{l}_n(\theta) - l(\theta)\right| \xrightarrow{a.s.} 0 \tag{27}$$

**Proof.** The corollary follows directly from the uniform law of large numbers. $\qquad\square$

## C  CONSISTENCY PROOFS

**Lemma 1** *Suppose for some $\epsilon > 0$ there exists a constant $B_p^{(\epsilon)}$ such that*

$$\int |\log \hat{f}_\theta(z)|^{1+\epsilon} p(z) dz \leq B_p^{(\epsilon)} < \infty \quad \forall \theta \in \Theta \tag{28}$$

*and there exists an $n_0$ such that for all $n > n_0$ there exists a constant $B_{\hat{q}}^{(\epsilon)}$ such that*

$$\int |\log \hat{f}_\theta(z)|^{1+\epsilon} \hat{q}_n^{(h_n)}(z) dz \leq B_{\hat{q}}^{(\epsilon)} < \infty \quad \forall \theta \in \Theta \tag{29}$$

*almost surely. Then, the inequality*

$$\sup_{\theta \in \Theta} \left| l_n^{(h)}(\theta) - l(\theta) \right| \leq C_\epsilon \left( \int |\hat{q}_n^{(h)}(z) - p(z)| dz \right)^{\frac{\epsilon}{1+\epsilon}} \tag{30}$$

*where $C_\epsilon$ is a constant holds with probability 1 for all $n > n_0$.*

**Proof of Lemma 1** Using Hoelder's inequality and the nonnegativity of $p$ and $\hat{q}_n^{(h)}$, we obtain

$$\left| l_n^{(h)}(\theta) - l(\theta) \right| = \left| \int \log \hat{f}_\theta(z)(\hat{q}_n^{(h)}(z) - p(z)) dz \right|$$

$$\leq \int |\log \hat{f}_\theta(z)| \, |\hat{q}_n^{(h)}(z) - p(z)| dz$$

$$= \int |\log \hat{f}_\theta(z)| \, |\hat{q}_n^{(h)}(z) - p(z)|^{\frac{1}{1+\epsilon}} \, |\hat{q}_n^{(h)}(z) - p(z)|^{\frac{\epsilon}{1+\epsilon}} dz$$

$$\leq \left( \int |\log \hat{f}_\theta(z)|^{1+\epsilon} \, |\hat{q}_n^{(h)}(z) - p(z)| dz \right)^{\frac{1}{1+\epsilon}}$$

$$\left( \int |\hat{q}_n^{(h)}(z) - p(z)| dz \right)^{\frac{\epsilon}{1+\epsilon}}$$

$$\leq \left( \int |\log \hat{f}_\theta(z)|^{1+\epsilon} \hat{q}_n^{(h)}(z) dz \right.$$

$$\left. + \int |\log \hat{f}_\theta(z)|^{1+\epsilon} p(z) dz \right)^{\frac{1}{1+\epsilon}} \left( \int |\hat{q}_n^{(h)}(z) - p(z)| dz \right)^{\frac{\epsilon}{1+\epsilon}}$$

Employing the regularity conditions (28) and (29) and writing $C^{(\epsilon)} = B_p^{(\epsilon)} + B_{\hat{q}}^{(\epsilon)}$, it follows that $\exists n_0$ such that $\forall n > n_0$

$$\sup_{\theta \in \Theta} \left| l_n^{(h)}(\theta) - l(\theta) \right| \leq \left( B_p^{(\epsilon)} + B_{\hat{q}}^{(\epsilon)} \right) \left( \int |\hat{q}_n^{(h)}(z) - p(z)| dz \right)^{\frac{\epsilon}{1+\epsilon}}$$

$$= C^{(\epsilon)} \left( \int |\hat{q}_n^{(h)}(z) - p(z)| dz \right)^{\frac{\epsilon}{1+\epsilon}}$$

with probability 1. □

Lemma 1 states regularity conditions ensuring that the expectations in $l_n^{(h)}(\theta)$ and $l(\theta)$ are well-behaved in the limit. In particular, (28) and (29) imply uniform and absolute integrability of the log-likelihoods under the respective probability measures induced by $p$ and $\hat{q}_n^{(h)}$. Since we are interested in the asymptotic behavior, it is sufficient for (29) to hold for $n$ large enough with probability 1.

Inequality (30) shows that we can make $|l_n^{(h)}(\theta) - l(\theta)|$ small by reducing the $l_1$-distance between the true density $p$ and the kernel density estimate $\hat{q}_n^{(h)}$. There exists already a vast body of literature, discussing how to properly choose the kernel $K$ and the bandwidth sequence $(h_n)$ so that $\int |\hat{q}_n^{(h_n)}(z) - p(z)| dz \to 0$. We employ the results in Devroye (1983) for our purposes, leading us to Proposition 1.

**Proof of Proposition 1.** Let $A$ denote the event that $\exists n_0 \forall n > n_0$ inequality (30) holds for some constant $C^{(\epsilon)}$. From our regularity assumptions it follows that $\mathbb{P}(A^c) = 0$. Given that $A$ holds, we just have to show that $\int |\hat{q}_n^{(h)}(z) - p(z)| dz \xrightarrow{a.s.} 0$. Then, the upper bound in (30) tends to zero and we can conclude our proposition.

For any $\delta > 0$ let $B_n$ denote the event

$$\int |\hat{q}_n^{(h)}(z) - p(z)| dz \leq \delta \tag{31}$$

wherein $\hat{q}_n^{(h)}(z)$ is a kernel density estimate obtained based on $n$ samples from $p(z)$. Under the conditions in (11) we can apply Theorem 1 of Devroye (1983), obtaining an upper bound on the probability that (31) does not hold, i.e. $\exists u, m_0$ such that $\mathbb{P}(B_n^c) \leq e^{-un}$ for all $n > m_0$.

Since we need both $A$ and $B_n$ for $n \to \infty$ to hold, we consider the intersection of the events $(A \cap B_n)$. Using a union bound argument it follows that $\exists k_0$ such that $\forall n > k_0 : \mathbb{P}((A \cap B_n)^c) \leq \mathbb{P}(A^c) + \mathbb{P}(B_n^c) = 0 + e^{-un} = e^{-un}$. Note that we can simply choose $k_0 = \max\{n_0, m_0\}$ for this to hold. Hence, $\sum_{n=k_0+1}^{\infty} P((A \cap B_n)^c) < \sum_{n=1}^{\infty} e^{-un} = \frac{1}{e^u - 1} < \infty$ and by the Borel-Cantelli lemma we can conclude that

$$\lim_{n \to \infty} \sup_{\theta \in \Theta} \left| l_n^{(h)}(\theta) - l(\theta) \right| = 0 \tag{32}$$

holds with probability 1. $\qquad\square$

**Proof of Theorem 1.** The inequality in (10) implies that for any $n > n_0$,

$$\varlimsup_{r \to \infty} \sup_{\theta \in \Theta} \left| \hat{l}_{n,r}^{(h)}(\theta) - l(\theta) \right| \leq \varlimsup_{r \to \infty} \sup_{\theta \in \Theta} \left| \hat{l}_{n,r}^{(h)}(\theta) - l_n^{(h)}(\theta) \right| + \sup_{\theta \in \Theta} \left| l_n^{(h)}(\theta) - l(\theta) \right| \tag{33}$$

Let $n > n_0$ be fixed but arbitrary and denote

$$J_{n,r} = \sup_{\theta \in \Theta} \left| \hat{l}_{n,r}^{(h)}(\theta) - l_n^{(h)}(\theta) \right| \quad r \in \mathbb{N}, n > n_0 \tag{34}$$

It is important to note that $J_{n,r}$ is a random variable that depends on the samples $\mathbf{Z}^{(n)} = (Z_1, ..., Z_n)$ as well as on the randomness inherent in Algorithm 1. We define $\mathbf{I}^{(r)} = (I_1, ... I_r)$ as the indices sampled uniformly from $\{1, ..., n\}$ and $\Xi^{(r)} = (\xi_1, ... xi_r)$ as the sequence of perturbation vectors sampled from $K$. Let $P(\mathbf{Z}^{(n)})$, $P(\mathbf{I}^{(r)})$ and $P(\Xi^{(r)})$ be probability measures of the respective random sequences.

If we fix $\mathbf{Z}^{(n)}$ to be equal to an arbitrary sequence $Z^{(n)}$, then $\hat{q}_n^{(h)}$ is fixed and we can treat $J_{n,r}$ as the regular difference between a sample estimate and expectation under $\hat{q}_n^{(h)}$. By the regularity condition (29), the compactness of $\Theta$ and the continuity of $f_\theta$ in $\theta$, we can invoke the uniform law of large numbers to show that

$$\lim_{r \to \infty} J_{n,r} = \lim_{r \to \infty} \sup_{\theta \in \Theta} \left| \hat{l}_{n,r}^{(h)}(\theta) - l_n^{(h)}(\theta) \right| = 0 \tag{35}$$

with probability 1.

Now we want to show that (35) also holds with probability 1 for random training samples $\mathbf{Z}^{(n)}$. First, we write $J_{n,r}$ as a deterministic function of random variables:

$$J_{n,r} = J(\mathbf{Z}^{(n)}, \mathbf{I}^{(r)}, \Xi^{(r)}) \tag{36}$$

This allows us to restate the result in (35) as follows:

$$\mathbb{P}_{\mathbf{I}^{(r)}, \Xi^{(r)}} \left( \forall \delta > 0 \, \exists r_0 \, \forall r > r_0 : J(\mathbf{Z}^{(n)} = Z^{(n)}, \mathbf{I}^{(r)}, \Xi^{(r)}) < \delta \right)$$
$$= \int \int \mathbf{1} \left( \forall \delta > 0 \, \exists r_0 \, \forall r > r_0 : J(\mathbf{Z}^{(n)} = Z^{(n)}, \mathbf{I}^{(r)}, \Xi^{(r)}) < \delta \right) dP(\Xi^{(r)}) dP(\mathbf{I}^{(r)}) \tag{37}$$
$$= 1$$

In that $\mathbf{1}(A)$ denotes an indicator function which returns 1 if $A$ is true and 0 else. Next we consider the probability that the convergence in (35) holds for random $\mathbf{Z}^{(n)}$:

$$
\begin{aligned}
&\mathbb{P}_{\mathbf{Z}^{(n)}, \mathbf{I}^{(r)}, \Xi^{(r)}} \left( \forall \delta > 0 \ \exists r_0 \ \forall r > r_0 : J(\mathbf{Z}^{(n)}, \mathbf{I}^{(r)}, \Xi^{(r)}) < \delta \right) \\
&= \int \int \int \mathbf{1} \left( \forall \delta > 0 \ \exists r_0 \ \forall r > r_0 : J(\mathbf{Z}^{(n)}, \mathbf{I}^{(r)}, \Xi^{(r)}) < \delta \right) dP(\Xi^{(r)}) dP(\mathbf{I}^{(r)}) dP(\mathbf{Z}^{(n)}) \\
&= \int dP(\mathbf{Z}^{(n)}) \underbrace{\left( \int \int \mathbf{1} \left( \forall \delta > 0 \ \exists r_0 \ \forall r > r_0 : J(\mathbf{Z}^{(n)}, \mathbf{I}^{(r)}, \Xi^{(r)}) < \delta \right) dP(\Xi^{(r)}) dP(\mathbf{I}^{(r)}) \right)}_{=1} \\
&= 1
\end{aligned}
$$

Note that we can $dP(\mathbf{Z}^{(n)})$ move outside of the inner integrals, since $\mathbf{Z}^{(n)}$ is independent from $\mathbf{I}^{(r)}$ and $\Xi^{(r)}$. Hence, we can conclude that (35) also holds, which we denote as event $A$, with probability 1 for random training data.

From Proposition 1 we know, that

$$
\lim_{n \to \infty} \sup_{\theta \in \Theta} \left| l_n^{(h)}(\theta) - l(\theta) \right| = 0 \tag{38}
$$

with probability 1. We denote the event that (38) holds as $B$. Since $P(A^c) = P(B^c) = 0$, we can use a union bound argument to show that $P(A \cap B) = 1$. From (35) and (33) it follows that for any $n > n_0$,

$$
\overline{\lim_{r \to \infty}} \sup_{\theta \in \Theta} \left| \hat{l}_{n,r}^{(h)}(\theta) - l(\theta) \right| \leq \sup_{\theta \in \Theta} \left| l_n^{(h)}(\theta) - l(\theta) \right| \tag{39}
$$

with probability 1. Finally, we combine this result with (38), obtaining that

$$
\lim_{n \to \infty} \overline{\lim_{r \to \infty}} \sup_{\theta \in \Theta} \left| \hat{l}_{n,r}^{(h)}(\theta) - l(\theta) \right| = 0 \tag{40}
$$

almost surely, which concludes the proof. $\qquad \square$

**Proof of Theorem 2.** The proof follows the argument used in Theorem 1 of White (1989). In the following, we assume that (13) holds. From Theorem 1 we know that this is the case with probability 1. Respectively, we only consider realizations of our training data $\mathbf{Z}^{(n)}$ and noise samples $\mathbf{I}^{(r)}, \Xi^{(r)}$ for which the convergence in (13) holds (see proof of Theorem 1 for details on this notation).

For such realization, let $(\hat{\theta}_{n,r}^{(h)})$ be minimizers of $\hat{l}_{n,r}^{(h)}$. Also let $(n_i)_i$ and for any $i$, $(r_{i,j})_j$ be increasing sequences of positive integers. Define $v_{i,j} := \hat{\theta}_{n_i, r_{i,j}}^{(h)}$ and $\mu_{i,j}(\theta) := \hat{l}_{n_i, r_{i,j}}^{(h)}(\theta)$. Due to the compactness of $\Theta$ and the Bolzano-Weierstrass property thereof, there exists a limit point $\theta^0 \in \Theta$ and increasing subsequences $(i_k)_k, (j_k)_k$ so that $v_{i_k, j_k} \to \theta^0$ as $k \to \infty$.

From the triangle inequality, it follows that for any $\epsilon > 0$ there exists $k_0$ so that $\forall k > k_0$

$$
|\mu_{i_k, j_k}(v_{i_k, j_k}) - l(\theta^0)| \leq |\mu_{i_k, j_k}(v_{i_k, j_k}) - l(v_{i_k, j_k})| + |l(v_{i_k, j_k}) - l(\theta^0)| < 2\epsilon \tag{41}
$$

given the convergence established in Theorem 1 and the continuity of $l$ in $\theta$. Next, the result above is extended to

$$
l(\theta^0) - l(\theta) = [l(\theta^0) - \mu_{i_k, j_k}(v_{i_k, j_k})] + [\mu_{i_k, j_k}(v_{i_k, j_k}) - \mu_{i_k, j_k}(\theta)] + [\mu_{i_k, j_k}(\theta) - l(\theta)] \leq 3\epsilon \tag{42}
$$

which again holds for $k$ large enough. This due to (41), $\mu_{i_k, j_k}(v_{i_k, j_k}) - \mu_{i_k, j_k}(\theta) \leq 0$ since $v_{i_k, j_k}$ is the minimizer of $\mu_{i_k, j_k}$, and $\mu_{i_k, j_k}(\theta) - l(\theta) < \epsilon$ by Theorem 1. Because $\epsilon$ can be made arbitrarily small, $l(\theta^0) \leq l(\theta)$ as $k \to \infty$. Because $\theta \in \Theta$ is arbitrary, $\theta^0$ must be in $\Theta^*$. In turn, since $(n_i)_i$, $(r_{i,j})_j$ and $(i_k)_k, (j_k)_k$ were chosen arbitrarily, every limit point of a sequence $(v_{i_k, j_k})_k$ must be in $\Theta^*$.

In the final step, we proof the theorem by contradiction. Suppose that (14) does not hold. In this case, there must exist an $\epsilon > 0$ and sequences $(n_i)_i, (r_{i,j})_j$ and $(i_k)_k, (j_k)_k$ such that $||(v_{i_k, j_k})_k - \bar{\theta}||_2 > \epsilon$ for all $k$ and $\bar{\theta} \in \Theta^*$. However, by the previous argument the limit point of the any sequence $(v_{i_k, j_k})_k$ must be in $\Theta^*$. That is a contradiction to $||(v_{i_k, j_k})_k - \bar{\theta}||_2 > \epsilon \ \forall \ k, \bar{\theta} \in \Theta^*$. Since

the random sequences $\mathbf{Z}^{(n)}, \mathbf{I}^{(r)}, \Xi^{(r)}$ where chosen from a set with probability mass of 1, we can conclude our proposition that

$$\lim_{n\to\infty} \overline{\lim_{r\to\infty}} \, d(\hat{\theta}_{n,r}^{(h)}, \Theta^*) = 0$$

almost surely. $\square$

**Discussion of Theorem 2.** Note that, similar to $\theta^*$, $\hat{\theta}_{n,r}^{(h)}$ does not have to be unique. In case there are multiple minimizers of $\hat{l}_{n,r}^{(h)}$, we can chose one of them arbitrarily and the proof of the theorem still holds. Theorem 2 considers global optimizers over a set of parameters $\Theta$, which may not be attainable in practical settings. However, the application of the theorem to the context of local optimization is straightforward when $\Theta$ is chosen as a compact neighborhood of a local minimum $\theta^*$ of $l$ (Holmstrom & Koistinen, 1992b). If we set $\Theta^* = \{\theta^*\}$ and restrict minimization over $\hat{l}_{n,r}^{(h)}$ to the local region, then $\hat{\theta}_{n,r}^{(h)}$ converges to $\Theta^*$ as $n, r \to \infty$ in the sense of Theorem 2.

# D    CONDITIONAL DENSITY ESTIMATION MODELS

## D.1    MIXTURE DENSITY NETWORK

Mixture Density Networks (MDNs) combine conventional neural networks with a mixture density model for the purpose of estimating conditional distributions $p(y|x)$ (Bishop, 1994). In particular, the parameters of the unconditional mixture distribution $p(y)$ are outputted by the neural network, which takes the conditional variable $x$ as input.

For our purpose, we employ a Gaussian Mixture Model (GMM) with diagonal covariance matrices as density model. The conditional density estimate $\hat{p}(y|x)$ follows as weighted sum of $K$ Gaussians

$$\hat{p}(y|x) = \sum_{k=1}^{K} w_k(x;\theta) \mathcal{N}\left(y|\mu_k(x;\theta), \sigma_k^2(x;\theta)\right) \tag{43}$$

wherein $w_k(x;\theta)$ denote the weight, $\mu_k(x;\theta)$ the mean and $\sigma_k^2(x;\theta)$ the variance of the k-th Gaussian component. All the GMM parameters are governed by the neural network with parameters $\theta$ and input $x$.

The mixing weights $w_k(x;\theta)$ must resemble a categorical distribution, i.e. it must hold that $\sum_{k=1}^{K} w_k(x;\theta) = 1$ and $w_k(x;\theta) \geq 0 \, \forall k$. To satisfy the conditions, the softmax linearity is used for the output neurons corresponding to $w_k(x;\theta)$. Similarly, the standard deviations $\sigma_k(x)$ must be positive, which is ensured by a sofplus non-linearity. Since the component means $\mu_k(x;\theta)$ are not subject to such restrictions, we use a linear output layer without non-linearity for the respective output neurons.

For the experiments in 5.2 and 5.1, we set $K = 10$ and use a neural network with two hidden layers of size 32.

### D.1.1    KERNEL MIXTURE NETWORK

While MDNs resemble a purely parametric conditional density model, a closely related approach, the Kernel Mixture Network (KMN), combines both non-parametric and parametric elements (Ambrogioni et al., 2017). Similar to MDNs, a mixture density model of $\hat{p}(y)$ is combined with a neural network which takes the conditional variable $x$ as an input. However, the neural network only controls the weights of the mixture components while the component centers and scales are fixed w.r.t. to $x$. For each of the kernel centers, $M$ different scale/bandwidth parameters $\sigma_m$ are chosen. As for MDNs, we employ Gaussians as mixture components, wherein the scale parameter directly coincides with the standard deviation.

Let $K$ be the number of kernel centers $\mu_k$ and $M$ the number of different kernel scales $\sigma_m$. The KMN conditional density estimate reads as follows:

$$\hat{p}(y|x) = \sum_{k=1}^{K} \sum_{m=1}^{M} w_{k,m}(x;\theta) \mathcal{N}(y|\mu_k, \sigma_m^2) \tag{44}$$

As previously, the weights $w_{k,m}$ correspond to a softmax function. The $M$ scale parameters $\sigma_m$ are learned jointly with the neural network parameters $\theta$. The centers $\mu_k$ are initially chosen by k-means clustering on the $\{y_i\}_{i=1}^n$ in the training data set. Overall, the KMN model is more restrictive than MDN as the locations and scales of the mixture components are fixed during inference and cannot be controlled by the neural network. However, due to the reduced flexibility of KMNs, they are less prone to over-fit than MDNs.

For the experiments in 5.2 and 5.1, we set $K = 50$ and $M = 2$. The respective neural network has two hidden layers of size 32.

### D.2 NORMALIZING FLOW NETWORK

The Normalizing Flow Network (NFN) is similar to the MDN and KMN in that a neural network takes the conditional variable $x$ as its input and outputs parameters for the distribution over $y$. For the NFN, the distribution is given by a Normalizing Flow (Rezende & Mohamed, 2015). It works by transforming a simple base distribution and an accordingly distributed random variable $Z_0$ through a series of invertible, parametrized mappings $f = f_N \circ \cdots \circ f_1$ into a successively more complex distribution $p(f(Z_0))$. The PDF of samples $\boldsymbol{z}_N \sim p(f(Z_0))$ can be evaluted using the change-of-variable formula:

$$\log p(\boldsymbol{z}_N) = \log p(\boldsymbol{z}_0) - \sum_{n=1}^{N} \log \left| \det \frac{\partial f_n}{\partial \boldsymbol{z}_{n-1}} \right| \tag{45}$$

The Normalizing Flows from Rezende & Mohamed (2015) were introduced in the context of posterior estimation in variational inference. They are optimized for fast sampling while the likelihood evaluation for externally provided data is comparatively slow. To make them useful for CDE, we invert the direction of the flows, defining a mapping from the transformed distribution $p(Z_N)$ to the base distribution $p(Z_0)$ by setting $\hat{f}_i^{-1}(\boldsymbol{z}_i) = f_i(\boldsymbol{z}_i)$.

We experimented with three types of flows: planar flows, radial flows as parametrized by Trippe & Turner (2018) and affine flows $f^{-1}(\boldsymbol{z}) = \exp(a)\boldsymbol{z} + b$. We have found that one affine flow combined with multiple radial flows performs favourably in most settings.

For the experiments in 5.2 and 5.1, we used a standard Gaussian as the base distribution that is transformed through one affine flow and ten radial flows. The respective neural network has two hidden layers of size 32.

### E SIMULATED DENSITIES AND DATASETS

### E.1 SKEWNORMAL

The data generating process $(x, y) \sim p(x, y)$ resembles a bivariate joint-distribution, wherein $x \in \mathbb{R}$ follows a normal distribution and $y \in \mathbb{R}$ a conditional skew-normal distribution (Anděl et al., 1984). The parameters $(\xi, \omega, \alpha)$ of the skew normal distribution are functionally dependent on $x$. Specifically, the functional dependencies are the following:

$$x \sim \mathcal{N}\left( \ \cdot \ \bigg| \mu = 0, \sigma = \frac{1}{2} \right) \tag{46}$$

$$\xi(x) = a * x + b \qquad a, b \in \mathbb{R} \tag{47}$$

$$\omega(x) = c * x^2 + d \qquad c, d \in \mathbb{R} \tag{48}$$

$$\alpha(x) = \alpha_{low} + \frac{1}{1 + e^{-x}} * (\alpha_{high} - \alpha_{low}) \tag{49}$$

$$y \sim SkewNormal\big(\xi(x), \omega(x), \alpha(x)\big) \tag{50}$$

Accordingly, the conditional probability density $p(y|x)$ corresponds to the skew normal density function:

$$p(y|x) = \frac{2}{\omega(x)} \mathcal{N}\left( \frac{y - \xi(x)}{\omega(x)} \right) \Phi\left( \alpha(x) \frac{y - \xi(x)}{\omega(x)} \right) \tag{51}$$

In that, $\mathcal{N}(\cdot)$ denotes the density, and $\Phi(\cdot)$ the cumulative distribution function of the standard normal distribution. The shape parameter $\alpha(x)$ controls the skewness and kurtosis of the distribution.

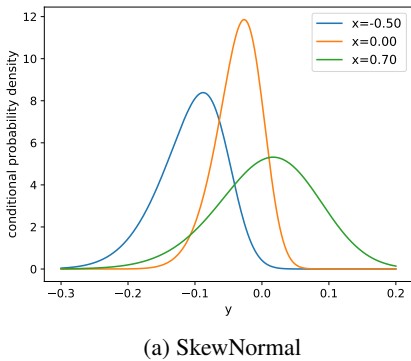
(a) SkewNormal

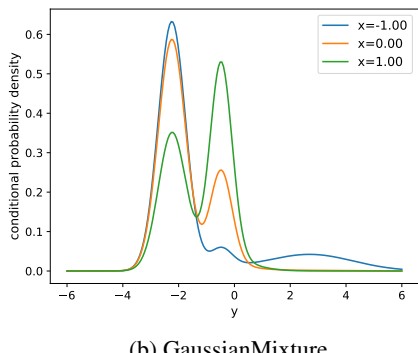
(b) GaussianMixture

Figure 4: **Conditional density simulation models.** Conditional probability densities corresponding to the different simulation models. The coloured graphs represent the probability densities $p(y|x)$, conditioned on different values of $x$.

We set $\alpha_{low} = -4$ and $\alpha_{high} = 0$, giving $p(y|x)$ a negative skewness that decreases as $x$ increases. This distribution will allow us to evaluate the performance of the density estimators in presence of skewness, a phenomenon that we often observe in financial market variables. Figure 4a illustrates the conditional skew normal distribution.

### E.2 GAUSSIAN MIXTURE

The joint distribution $p(x, y)$ follows a Gaussian Mixture Model in $\mathbb{R}^4$ with 5 Gaussian components, i.e. $K = 5$. We assume that $x \in \mathbb{R}^2$ and $y \in \mathbb{R}^2$ can be factorized, i.e.

$$p(x, y) = \sum_{i=1}^{K} w_k \, \mathcal{N}(y|\mu_{y,k}, \Sigma_{y,k}) \mathcal{N}(x|\mu_{x,k}, \Sigma_{x,k}) \tag{52}$$

When $x$ and $y$ can be factorized as in (52), the conditional density $p(y|x)$ can be derived in closed form:

$$p(y|x) = \sum_{i=1}^{K} W_k(x) \, \mathcal{N}(y|\mu_{y,k}, \Sigma_{y,k}) \tag{53}$$

wherein the mixture weights are a function of $x$:

$$W_k(x) = \frac{w_k \, \mathcal{N}(x|\mu_{x,k}, \Sigma_{x,k})}{\sum_{j=1}^{K} w_k \, \mathcal{N}(x|\mu_{x,j}, \Sigma_{x,j})} \tag{54}$$

For details and derivations we refer the interested reader to Guang Sung (2004) and Gilardi et al. (2002). The weights $w_k$ are sampled from a uniform distribution $U(0, 1)$ and then normalized to sum to one. The component means are sampled from a spherical Gaussian with zero mean and standard deviation of $\sigma = 1.5$. The covariance matrices $\Sigma_{y,k}$) and $\Sigma_{y,k}$) are sampled from a Gaussian with mean 1 and standard deviation 0.5, and then projected onto the cone of positive definite matrices.

Since we can hardly visualize a 4-dimensional GMM, Figure 4b depicts a 2-dimensional equivalent, generated with the procedure explained above.

### E.3 EURO STOXX 50 DATA

The Euro Stoxx 50 data comprises 3169 trading days, dated from January 2003 until June 2015. The goal is to predict the conditional probability density of 1-day log-returns, conditioned on 14 explanatory variables. These conditional variables comprise classical return factors from finance as well as option implied moments. For details, we refer to Rothfuss et al. (2019). Overall, the target variable is one-dimensional, i.e. $y \in \mathcal{Y} \subseteq \mathbb{R}$, whereas the conditional variable $x$ constitutes a 14-dimensional vector, i.e. $x \in \mathcal{X} \subseteq \mathbb{R}^{14}$.

### E.4 NYC TAXI DATA

We follow the setup in Dutordoir et al. (2018). The dataset contains records of taxi trips in the Manhattan area operated in January 2016. The objective is to predict spatial distributions of the drop-off location, based on the pick-up location, the day of the week, and the time of day. In that, the two temporal features are represented as sine and cosine with natural periods. Accordingly, the target variable $y$ is 2-dimensional (longitude and latitude of dropoff-location) whereas the conditional variable is 6-dimensional. From the ca. 1 million trips, we randomly sample 10,000 trips to serve as training data.

### E.5 UCI

**Boston Housing**  Concerns the value of houses in the suburban area of Boston. Conditional variables are mostly socio-economic as well as geographical factors. For more details see https://archive.ics.uci.edu/ml/machine-learning-databases/housing/

**Concrete**  The task is to predict the compressive strength of concrete given variables describing the conrete composition. For more details see https://archive.ics.uci.edu/ml/machine-learning-databases/concrete/compressive/

**Energy**  Concerns the energy efficiency of homes. The task is to predict the cooling load based on features describing the build of the respective house. For more details see https://archive.ics.uci.edu/ml/datasets/energy+efficiency

