# OpenReview forum: "Noise Regularization for Conditional Density Estimation"
_ICLR.cc/2020/Conference — Reject_

### Official Review · AnonReviewer3 · 2019-10-25
**Official Blind Review #3**

**Rating:** 6

**Review:**

The paper considers the problem of parametric conditional density estimation, i.e. given a set of points {(x_n, y_n)} drawn from a distribution $p(x,y)$, the task is to estimate the conditional distribution p(x|y). The paper considers parametric estimation where in given a parametrized family of distributions f_{theta} we wish to minimize the likelihood of seeing the given data over theta. The parametric family in a lot of applications consists of highly expressive families like neural networks, which leads to the issue of overfitting in small data regimes. This has been tackled via regularization over the parameter space which might be hard to interpret as the associated inductive bias is not well understood and depends on the parametric family under consideration. On the other hand the paper proposes to add explicit noise in the examples used during training, i.e. irrespective of the optimization procedure (which could be mini-bath sgd) the paper proposes to draw examples from the data set, explicitly add noise onto the examples and create a proxy objective over the augmented data set.

The paper establishes two theoretical results. First is a simple taylor approximation based analysis to highlight the effect of the variance of noise. The conclusion is that higher variance penalizes the high curvature areas of the resultant density and hence this kind of noise addition could be seen as making resulting density smoother. The second contribution is to show that this procedure is asymptotically consistent, i.e. as n goes to infinity and the number of augmented data points go to infinity, the resulting density converges to the target density. This of course requires the noise variance to follow a decreasing schedule to 0.

The main merit of the idea is in its agnostic nature, as it can be applied to any parametric family and the experiements show that it seems to uniform improvement across models,  The basic idea proposed by the error, has existed in the space of deep learning based methods forever. This is the same idea behind image augmentation which forms a crucial part of training supervised models in vision. The authors claim that this idea is novel in the space of parametric density estimation, however I do not know enough about the area to verify the claim. It would surprise that this very natural idea has not been tried before.

I have gone through the theoretical derivations in the paper and they look sound to me. However the results are all asymptotic in nature without establishing explicit rates which is a little bit of disappointment. Since I am not completely familiar in nature, but I guess such asymptotic consistency might be achievable using other forms of regularization under suitable assumptions. In that light, the theoretical contributions while being sound did not lend much intuition about why such a method might outperform others. Intuition does arise from the derivation for the effect of noise on the objective which helps understand the nature of the noise, but one wonders if similar intuitions could be derived for other forms of regularization as well. It would be great to see this derivation being extended to some concrete scenarios under well understood parametric families and seeing the effect explicitly.

Regarding the experiments - The experiments definitely look promising as the improvments seem uniformly good across the cases considered. I am not an expert however in this setting so it is hard for me to judge the quality and significance of the benchmarks. The experiment methodology nevertheless looks sound.


**Experience Assessment:**

I do not know much about this area.

**Review Assessment: Checking Correctness Of Derivations And Theory:**

I carefully checked the derivations and theory.

**Review Assessment: Checking Correctness Of Experiments:**

I assessed the sensibility of the experiments.

**Review Assessment: Thoroughness In Paper Reading:**

I read the paper at least twice and used my best judgement in assessing the paper.

---

> ### Author Response · Authors · 2019-11-11
> **Authors' response**
>
> We thank the reviewer for the detailed elaborations and helpful feedback w.r.t. the technical part of the paper. In the following, we attempt to address the reviewer's concerns:
>
> C: The authors claim that this idea is novel in the space of parametric density estimation [...]. It would surprise that this very natural idea has not been tried before.
>
> R: As stated in the paper, we are not aware of any work in the intersection of conditional density estimation and noise regularization. We welcome any reference in this area which we might have missed.
>
> C:  [...] Results are all asymptotic in nature without establishing explicit rates which is a little bit of disappointment. [...] In that light, the theoretical contributions while being sound did not lend much intuition about why such a method might outperform others.
>
> R: In this response, we assume that with “explicit rates” the reviewer refers to convergence rates for finite sample sizes n. We agree with the reviewer that finite sample bounds rather than asymptotic rates would be more desirable. However, the requirements on the smoothing rate which are a side product of Theorem 1 provide us with intuition on how to adjust the noise variance in response to the sample size and data dimensionality. Deriving finite sample bounds would require making limiting model assumptions which would be conflict with the model-agnostic feature and generality of this work.
>
> C: Intuition does arise from the derivation for the effect of noise on the objective which helps understand the nature of the noise, but one wonders if similar intuitions could be derived for other forms of regularization as well.
>
> R: The key property of noise regularization is its nonparametric nature which allows us to derive the regularization effect uniformly for all parametric CDE function classes. In contrast, if regularization targets the parameters of a model, obtaining such a general, model-agnostic result is no longer possible.

---

### Official Review · AnonReviewer1 · 2019-10-26
**Official Blind Review #1**

**Rating:** 3

**Review:**

This paper proposes a noise regularization method which adds noise on both x and y for conditional density estimation problem (e.g., regression and classification). The writing is good and the whole paper is easy to follow. However, I vote for reject, since the novelty is somehow limited, the claims made in the paper is not well supported and experiments are not very convincing.

1. Adding noise on x (e.g., [1]), y (e.g., [2]) is not new. Though it is claimed that this paper extends previous results on classification/regression to conditional density estimation which is a more general case. This claim is not well supported. Experiments are still evaluated in classification/regression tasks.

2. Theorem 1 & 2 in Sec 4.2 only show the asymptotic case, which are quite obvious and seems helpless in understanding the advantage of adding noise regularization in conditional density estimation.

3. Sec 4.1. The explanation that Page 5, ```"The second term in (6) penalizes large negative second derivatives of the conditional log density estimate...". It is hard for me to understand. Large positive second derivatives also lead to poor smoothness.

[1] Learning with Marginalized Corrupted Features, ICML 2013
[2] Learning with Noisy Labels, NIPS 2013

**Experience Assessment:**

I have read many papers in this area.

**Review Assessment: Checking Correctness Of Derivations And Theory:**

I assessed the sensibility of the derivations and theory.

**Review Assessment: Checking Correctness Of Experiments:**

I assessed the sensibility of the experiments.

**Review Assessment: Thoroughness In Paper Reading:**

I read the paper at least twice and used my best judgement in assessing the paper.

---

> ### Author Response · Authors · 2019-11-11
> **Authors' response**
>
> We thank the reviewer for the feedback. In the following, we attempt to address the reviewer's concerns:
>
> C: Adding noise on x (e.g., [1]), y (e.g., [2]) is not new. Though it is claimed that this paper extends previous results on classification/regression to conditional density estimation which is a more general case. This claim is not well supported. Experiments are still evaluated in classification/regression tasks.
>
> R: As pointed out in the introduction, the paper addresses the regularization of conditional density estimation (CDE). As we acknowledge in the last paragraph of chapter 3, noise regularization is a well studied method in the context of regression and classification problems. The references mentioned by the reviewer fall into this body of literature. We thank the reviewer for pointing out additional papers we have missed and added them to our manuscript.
>
> The reviewer correctly points out classification / regression are special cases of more general class of CDE. That is, the predictive distribution is typically a simple Gaussian / Softmax-Categorical distribution. The main difficulties arising from generalizing beyond these special cases are moving towards more complex, expressive density models (e.g. GMM or Normalizing Flow) that are conjoined with a neural network and dealing with the inherent unboundedness of the log probability in Theorem 1 & 2.
>
> There exists a substantial body of empirical results for noise regularization in regression and classification tasks. In contrast, our paper focuses on CDE with expressive density models and datasets with a large amount of aleatoric uncertainty, evaluated with the negative log-likelihood, which is a standard metric for assessing the fit of a CDE. We would highly appreciate specific suggestions on how to improve the experiments and substantiate the claims.
>
> C: Theorem 1 & 2 in Sec 4.2 only show the asymptotic case, which are quite obvious and seems helpless in understanding the advantage of adding noise regularization in conditional density estimation.
>
> R: We agree with the reviewer that finite sample bounds rather than asymptotic consistency results would be more desirable. Obtaining finite sample bounds requires narrow assumptions on the density model and thus would reduce the generality / applicability of the results.  Moreover, the requirements on the smoothing rate which are a product of Theorem 1 provide us with intuition on how to adjust the noise variance in response to the sample size and the dimensionality of the data.
>
> C: Sec 4.1: "The second term in (6) penalizes large negative second derivatives of the conditional log density estimate...". It is hard for me to understand. Large positive second derivatives also lead to poor smoothness.
>
> R: The first term in (6), that is the negative log likelihood, pushes the density towards high probabilities in the data points y_i. Due to the limited probability mass, this leads to large negative second derivatives in the respective point. This is illustrated in Fig. 1 (left).  At the same time the second term in (6) penalizes large negative second derivatives in the locations y_i which counteracts the effect of the first term. Since both effects compensate each other, the regularization does not lead to large positive derivatives in the data locations. We will improve the clarity of our exposition in our revised paper.

---

### Official Review · AnonReviewer2 · 2019-10-26
**Official Blind Review #2**

**Rating:** 3

**Review:**

The paper presents a regularization technique for conditional density estimation. The method is simple: adding noise to the data points, and training on the noisy data points. The paper also further gives an interpretation of the method, as a form of smoothing the curvature of the density function. It further proves the consistency of the method.

Pros:
(i) The paper is well written.
(ii) The method is simple to implement.
(iii) The authors demonstrate a clear intuition of what the method does, i.e., as a form of smoothing.

Cons:
(i)  The method itself is not novel. Adding noise to for regularization is a quite common technique used in many different applications.
(ii) The experiments performed in the paper are all very small scaled. Not very convincing.
(iii) Continuing the last point, it is unclear whether such technique can be scaled up. The method in another view is to replace the empirical distribution with a kernel density estimate. We know kernel methods don't scale. It's hence questionable whether the method can give any benefits in large scale dataset.
(iv) If the use of noise regularization is to smooth the density function, how does it compare to methods that enforces Lipschitz continuity? e.g., gradient penalty, or construct a lipschtz constrained network such as in [1].

[1] Sorting out lipschitz function approximation. Anil et al.



**Experience Assessment:**

I have published one or two papers in this area.

**Review Assessment: Checking Correctness Of Derivations And Theory:**

I assessed the sensibility of the derivations and theory.

**Review Assessment: Checking Correctness Of Experiments:**

I assessed the sensibility of the experiments.

**Review Assessment: Thoroughness In Paper Reading:**

I read the paper at least twice and used my best judgement in assessing the paper.

---

> ### Author Response · Authors · 2019-11-11
> **Authors' response**
>
> We thank the reviewer for the valuable feedback. In the following, we attempt to address the reviewer's concerns:
>
> Concern: The experiments performed in the paper are all very small scaled.
>
> Response: In the large scale setting, the superior performance of neural network estimators over non-parametric methods is widely acknowledged. In contrast, the focus of the paper are small scale settings where data is scarce and classical nonparametric methods are considered state-of-the-art [1, 2]. One of the core contributions of the paper is the empirical study in section 5.3 of the paper, which shows that neural network based CDE, when combined with noise regularization, consistently & significantly outperforms previous non/semi-parametric methods, even when only very little training data is available.
>
> C: The method itself is not novel. Adding noise to for regularization is a quite common technique used in many different applications.
>
> R: We agree with the reviewer: Noise regularization has widely been studied in the context of regression and classification. The paper attempts to close a gap in previous research by contributing a comprehensive study of noise regularization for conditional density estimation, on which we could not find any previous work. Existing work on noise regularization for regression / classification (e.g. [3, 4]) typically assumes a simple Gaussian / Softmax as predictive distribution, leading to simple interactions between NN outputs and the resulting distribution which can be exploited for technical analysis. In contrast, our analysis makes no such assumptions, thus requiring non-trivial technical arguments to deal with the inherent unboundedness the log-probability and integrability issues, making it much more general. We will try  to further clarify the contribution in the paper’s introduction.
>
> C: [...] How does it compare to methods that enforce Lipschitz continuity? e.g., gradient penalty, or construct a lipschitz constrained network
>
> R: As is discussed in chapter 4 of the paper, the effect of regularization in the weight space of the neural network (NN) is only poorly understood in the context of CDE, since the NN only indirectly controls the prediction output through a complex density model. We have not considered further methods such as enforcing Lipschitz continuity as, in addition to restrictions to the NN weights and nonlinearities, this would also require imposing restrictions on the density model (e.g. Mixture Model  / Normalizing Flow). The non-parametric / model-agnostic nature of noise regularization circumvents such problems and makes it nearly universally applicable.
>
> [1] Masashi Sugiyama and Ichiro Takeuchi. Conditional density estimation via Least-Squares Density Ratio Estimation. AISTATS, 2010
> [2] Gaussian Process Conditional Density Estimation. NeurIPS, 2018
> [3] Holmstrom and P. Koistinen.  Using additive noise in back-propagation training. IEEE Transactions on Neural Networks, 1992
> [4] Learning with Marginalized Corrupted Features, ICML 2013

---

### Decision · Program_Chairs · 2019-12-19

**Decision:**

Reject

**Comment:**

This paper has a mixture of weak reviews, the majority of which lean towards reject. All reviews mention a lack of novelty, and 2 of 3 a lack of support in experiments. While the authors argue, perhaps legitimately, for the novelty of the paper with respect to current literature, this is not convincing in the exposition. I recommend that the authors improve the justification for the novelty of their methodology, and strengthen the experiments to convince reviewers. As it stands, this paper is not quite ready for publication.